# Efficient Graph Similarity Computation with Alignment Regularization

**Wei Zhuo**
Shenzhen Campus of Sun Yat-sen University
zhuow5@mail2.sysu.edu.cn

**Guang Tan**
Shenzhen Campus of Sun Yat-sen University
tanguang@mail.sysu.edu.cn

## Abstract

We consider the graph similarity computation (GSC) task based on graph edit distance (GED) estimation. State-of-the-art methods treat GSC as a learning-based prediction task using Graph Neural Networks (GNNs). To capture fine-grained interactions between pair-wise graphs, these methods mostly contain a node-level matching module in the end-to-end learning pipeline, which causes high computational costs in both the training and inference stages. We show that the expensive node-to-node matching module is not necessary for GSC, and high-quality learning can be attained with a simple yet powerful regularization technique, which we call the *Alignment Regularization* (AReg). In the training stage, the AReg term imposes a node-graph correspondence constraint on the GNN encoder. In the inference stage, the graph-level representations learned by the GNN encoder are directly used to compute the similarity score without using AReg again to speed up inference. We further propose a multi-scale GED discriminator to enhance the expressive ability of the learned representations. Extensive experiments on real-world datasets demonstrate the effectiveness, efficiency and transferability of our approach.

## 1 Introduction

Graph similarity computation (GSC) is a fundamental task in graph databases and plays a critical role in many real-world applications, including drug design [14, 28], program analysis [16], and social group identification [21, 27]. For example, one can search a drug database for a query chemical compound, in order to identify drugs with high similarity in structures or attributes and thus similar curative effects as desired [23]. To measure the similarity between pair-wise graphs, Graph Edit Distance (GED) [5] has been a major metric due to its generality, and many other graph similarity measures have been proven to be its special cases [17]. Unfortunately, computing exact GED is an NP-hard problem in general [12].

With the provably expressive power in distinguishing graph structures [32, 19, 7, 34], Graph Neural Networks (GNNs) have been adopted for GED approximation and shown to achieve superior performance on accuracy. Most state-of-the-art GNN-based GSC models [1, 3, 8, 18, 16] contain two sequential submodules in the end-to-end learnable pipeline (left of Fig. 1): (1) a GNN encoder, which is shared across two graphs to embed nodes into representation vectors to capture the intra-graph structure and feature information; (2) a matching model, which computes cross-graph node-level similarity, i.e., how a node in one graph relates to all the nodes in the other graph. The model outputs a summarized vector that fuses the node-level similarities between two graphs. Then, the similarity score is predicted based on the summarized vector via a regression head. The computational cost of such a sequential framework mainly comes from the matching model, which requires computational and memory cost quadratic in the number of nodes and sometimes involves additional parameters such as attention weights [18, 16], leading to heavy time consumption in both the training and inference

stages. Especially in the inference stage, we need to query every testing graph from the database. The recent approach EGSC [22] speeds up the similarity learning by dropping the matching model from SimGNN [1]. However, since cross-graph node-level interactions are ignored, EGSC cannot capture finer-grained similarity information, and thus results in suboptimal prediction performance. To overcome the intrinsic tension between predictive accuracy and speed, we propose a separated neural structure (right of Fig. 1) that detaches the matching model from the sequential pipeline, where the matching model only acts as a regularization term in the training stage to help the GNN encoder capture fine-grained similarity information, while in the inference (testing) stage, since the latent cross-graph interactions have been learned by the GNN encoder, the final similarity score can be directly computed based on the output representations of the GNN encoder without invoking the matching model again.

We show that explicitly learning the cross-graph node-to-node similarity is unnecessary, as the correlation information contained in the features and graph topology, when properly exploited, is sufficient to reflect such cross-graph interactions (see Section 3). Specifically, the problem of GED computation is equivalent to finding an optimal permutation such that the adjacency matrices of the two graphs can be best aligned. Hence, by analyzing the necessary conditions under the optimal permutation, we find that the best matching between two graphs can be inferred by minimizing the difference between the *intra-graph node-graph similarity* and *cross-graph node-graph similarity*. It motivates us to design a task-agnostic matching model based on the input data itself, with a novel regularization technique, called the Alignment

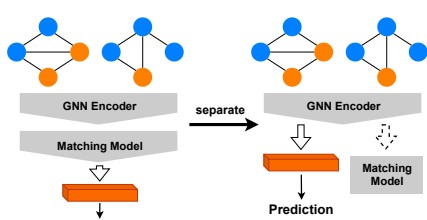

Figure 1: Illustration of separating the matching model from the end-to-end GSC framework to achieve a fast model (right side). In the fast model, the dotted arrow means the matching model does not participate in the similarity computation in the inference stage.

Regularization (AReg). AReg obviates the need of a matching model in the inference stage, thus making the model more efficient. AReg is also model-agnostic and can be applied to other GNN-based GSC models.

On the other hand, GNN-based GSC models usually use a single GED discriminator followed by a regression head to fuse graph representations from pair-wise input graphs and output a final similarity prediction score. We show that a single GED discriminator does not fully capture the dissimilarity between two graphs, while diverse discriminators may provide complementary information to reflect GED more accurately. Thus, we propose a multi-scale GED discriminator to improve the discriminability of the learned representations. We call the overall framework ERIC: Efficient gRaph sImilarity Computation, and conduct extensive experiments to verify the efficacy of our design. Results on several real-world datasets demonstrate that ERIC achieves state-of-the-art performance by significantly outperforming the baselines.

## 2  Preliminary

**Problem Formulation of GSC.**  Given a graph database $\mathcal{D}$ and a set of query graphs $\mathcal{Q}$, the problem of graph similarity computation is to produce a similarity score $\boldsymbol{y}$ between $\forall G_i \in \mathcal{Q}$ and $\forall G_j \in \mathcal{D}$, i.e., $\boldsymbol{y} = s(G_i, G_j)$ where $s : \mathcal{D} \times \mathcal{Q} \to \mathbb{R}^{(0,1]}$ is a similarity estimator. A graph $G \in \mathcal{D} \cup \mathcal{Q}$ is defined as $G = (V, E)$, where $V = \{v_k\}_{k=1}^N$ is a node set and $E \in V \times V$ is an edge set. In our setting, all the accessible graphs are unweighted and undirected. $V$ and $E$ jointly formulate an adjacency matrix $\boldsymbol{A} \in \mathbb{R}^{N \times N}$. If nodes are accompanied by features (e.g., labels or attribute vectors), they are represented as $\boldsymbol{X} \in \mathbb{R}^{N \times d}$ with dimension $d$.

**Graph Edit Distance (GED).**  GED as a graph similarity measure has been popularly adopted on graph search queries, due to its capacity to capture the structural and feature differences between graphs. As shown in Fig. 2, GED is defined as the number of edit operations in the optimal path that transforms $G_i$ into $G_j$, where the possible edit operations under consideration include edge

deletion/insertion, node deletion[1]/insertion, and node relabeling. To well fit the end-to-end regression task, instead of directly estimating the GED between two graphs, we convert the GED to the ground truth similarity score that the learning model aims to approximate. Following [1], the normalized GED is defined as $\text{nGED}(G_i, G_j) = \frac{\text{GED}(G_i, G_j)}{(|V_i| + |V_j|)/2}$, and the ground truth similarity score between $G_i$ and $G_j$ is defined as the normalized exponential of GED, resulting in a value ranging $(0, 1]$, i.e., $\mathbf{S}_{ij} = \exp(-\text{nGED}(G_i, G_j))$.

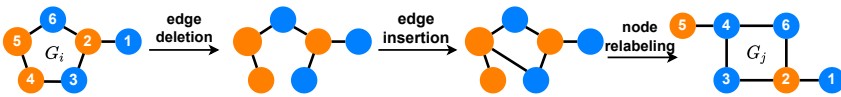

Figure 2: The optimal edit path with 3 edit operations to transform $G_i$ to $G_j$. As a result, $\text{GED}(G_i, G_j) = 3$.

## 3 Motivation: Analyzing GED in Embedding Space

Given two graphs $G_i = (\boldsymbol{A}_i, \boldsymbol{X}_i)$ and $G_j = (\boldsymbol{A}_j, \boldsymbol{X}_j)$ with the same number of nodes $N$ (If they have different numbers of nodes, pad the smaller $\boldsymbol{A}$ and $\boldsymbol{X}$ with zeros to make the two graphs equal in size), $\pi(\cdot)$ is a node index permutation that preserves the adjacency matrix. $\pi(\boldsymbol{A})$ denotes the adjacency matrix after node index permuting. We divide the computation of GED between $G_i$ and $G_j$ into two steps: (i) finding a permutation for $G_j$, such that

$$c = \min_\pi \sum_{k,l} \left| (\boldsymbol{A}_i - \pi(\boldsymbol{A}_j))[k,l] \right|, \tag{1}$$

where $c$ is the sum of the absolute values of all elements in matrix $\boldsymbol{A}_i - \pi(\boldsymbol{A}_j)$. We denote the optimal permutation satisfying Eq. (1) as $\pi^\star$. (ii) counting the number of cross-graph node pairs with the same indices yet different features, denoted as $m$. Then $\text{GED}(G_i, G_j) = \frac{c}{2} + m$. Obviously, if two graphs are topologically isomorphic, there exists $\pi = \pi^\star$ such that $\boldsymbol{A}_i = \pi^\star(\boldsymbol{A}_j)$, i.e., $c = 0$. Hence the GED is only determined by distinct features. For another example, when the node permutation $\pi$ assigns indices to $G_j$ as shown in Fig. 2, the structure of $G_i$ and $G_j$ can be best aligned, i.e., $c = 4$. Then, only one pair of nodes with the same index across graphs have different features (e.g., node 4). Thus, $\text{GED}(G_i, G_j) = 3$. The core of this two-step method is finding the optimal permutation to best align two graphs, then $m$ is determined under such alignment. Traversing all possible permutations to find $\pi^\star$ is also NP-hard, so we make some heuristic rules from Eq. (1) to guide the model design. Under the optimal permutation $\pi^\star$, the row similarity between $\boldsymbol{A}_i$ and $\pi^\star(\boldsymbol{A}_j)$ is maximized, so given an injective function $f_\theta(\cdot) : \mathbb{R}^N \to \mathbb{R}^d$ parameterized by $\theta$ to guarantee that nodes with different connectivity can be distinguished, a necessary condition is that the distance between $f_\theta(\boldsymbol{A}_i[k])$ and $f_\theta(\pi^\star(\boldsymbol{A}_j)[k])$ is minimized for all $k \in \{1, \cdots, N\}$. From a global view, the matrix similarity between $\boldsymbol{A}_i$ and $\pi^\star(\boldsymbol{A}_j)$ is also maximized. Given an injective function $g_\phi(\cdot) : \mathbb{R}^{N \times N} \to \mathbb{R}^d$ parameterized by $\phi$, another necessary condition under the optimal permutation is that the distance between $g_\phi(\boldsymbol{A}_i)$ and $g_\phi(\pi^\star(\boldsymbol{A}_j))$ is minimized. Thus, the optimal permutation $\pi^\star$ in Eq. (1) also satisfies the following function,

$$\pi^\star = \arg\min_\pi \text{DIST}\left(f_\theta(\boldsymbol{A}_i[k]), f_\theta(\pi(\boldsymbol{A}_j)[k])\right) + \text{DIST}\left(g_\phi(\boldsymbol{A}_i), g_\phi(\pi(\boldsymbol{A}_j))\right) \quad \forall k \in \{1, \cdots, N\}, \tag{2}$$

where $\text{DIST}(\cdot, \cdot)$ is a distance metric. The two terms in Eq. (2) can reflect GED at node-level and global-level respectively. Further, we regard $\{f_\theta(\boldsymbol{A}_i[k])\}_{k=1}^N \cup \{f_\theta(\pi^\star(\boldsymbol{A}_j)[k])\}_{k=1}^N$ as $2N$ anchors and assume $N \gg d$. When the second term of Eq. (2) reaches a minimum, it indicates that $g_\phi(\boldsymbol{A}_i)$ and $g_\phi(\pi^\star(\boldsymbol{A}_j))$ have similar distances to all anchors, i.e., the following $\gamma_i$ and $\gamma_j$ take the minimum value when $\pi = \pi^\star$,

$$\gamma_i = \sum_k^N \|\text{DIST}\left(f_\theta(\boldsymbol{A}_i[k]), g_\phi(\boldsymbol{A}_i)\right) - \text{DIST}\left(f_\theta(\boldsymbol{A}_i[k]), g_\phi(\pi(\boldsymbol{A}_j))\right)\|_2 \tag{3}$$

---

[1]For node deletion, all edges connected to the deleted node are also deleted. Although editing of multiple edges is involved, it is a single-time effort. Thus, node deletion is treated as a single graph edit operation.

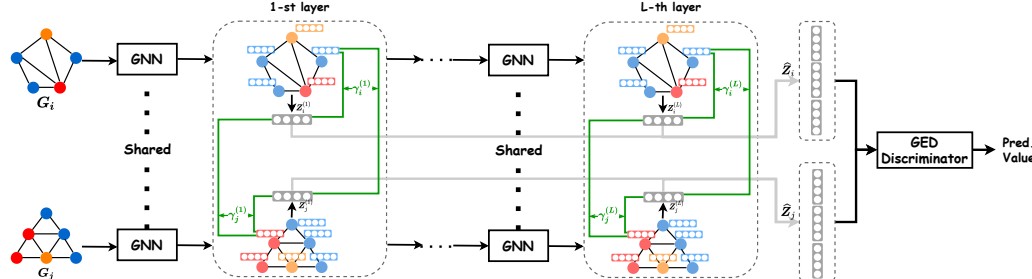

Figure 3: Overview of ERIC. The GNN encoder is shared across two graphs. The green lines denote AReg. The summarized graph representations $\widehat{\boldsymbol{Z}}_i$ and $\widehat{\boldsymbol{Z}}_j$ are combinations of graph representations learned in each layer, which are fed into the GED discriminator followed by a regression function to obtain the prediction value. In the inference stage, the AReg submodule is removed.

$$\gamma_j = \sum_k^N \left\| \text{DIST}\left(f_\theta(\pi(\boldsymbol{A}_j)[k]), g_\phi(\boldsymbol{A}_i)\right) - \text{DIST}\left(f_\theta(\pi(\boldsymbol{A}_j)[k]), g_\phi(\pi(\boldsymbol{A}_j))\right) \right\|_2. \tag{4}$$

On the other hand, the first term of Eq. (2) is a finer-grained alignment between two graphs, and $\pi^\star$ causes the first term of Eq. (2) to reach a minimum. Since the only difference between $\gamma_i$ and $\gamma_j$ is that $\gamma_j$ replaces $f_\theta(\boldsymbol{A}_i[k])$ in $\gamma_i$ with $f_\theta(\pi(\boldsymbol{A}_j)[k])$, the distance between $f_\theta(\boldsymbol{A}_i[k])$ and $f_\theta(\pi(\boldsymbol{A}_j)[k])$ can be reflected by the difference between $\gamma_i$ and $\gamma_j$, i.e., $\|\gamma_i - \gamma_j\|_2$. Hence, under the optimal permutation $\pi = \pi^\star$, $G_i$ and $G_j$ are best aligned, and so Eq. (2) is established, which is equivalent to $\gamma_i$, $\gamma_j$, and $\|\gamma_i - \gamma_j\|_2$ all taking the minimum values. These terms reflect similarities between pair-wise graphs at both node- and graph- levels, which can be inferred from the graph structure itself without consulting the ground-truth GEDs. It motivates us to separate the matching model from the end-to-end pipeline. In other words, instead of directly searching the optimal permutation $\pi^\star$, we derive the necessary conditions under $\pi^\star$, which impose additional constraints on the coordinates of nodes in the embedding space. These conditions are not necessarily sufficient, nevertheless they can provide useful knowledge for the model to learn representations that are better tailored to the GSC task. In Appendix A, we further analyze the rationality of this approximation by analyzing the sufficiency of these conditions. Assuming $g_\phi$ is a permutation-invariant function, then all permutation operators in Eq. (3) and Eq. (4) can be removed. Moreover, instead of computing cross-graph node-to-node similarity to compute a 'soft' alignment as done in most related work, $\gamma_i$ and $\gamma_j$ only compute node-to-graph similarity, which is more efficient.

## 4    Proposed Model: ERIC

The proposed ERIC framework mainly consists of two submodules: the Alignment Regularization (AReg) module and the Multi-Scale GED Discriminator, as depicted in Fig. 3. AReg aims to train the shared GNN encoder to enable the GNN to capture underlying alignment information between pair-wise graphs. The multi-scale GED discriminator trains the same GNN encoder so that the learned representations of two graphs can accurately reflect the GED. Such a paradigm follows the intuition of the GED definition that when two graphs are best aligned, the difference between them is the GED.

### 4.1    Alignment Regularization

Learning the optimal alignment between graphs is crucial for GED estimation, however, most existing GSC models rely on intricate cross-graph node-to-node similarity computation to learn a 'soft' alignment. Furthermore, learning node-to-node similarity inevitably yields a dense similarity matrix, which could introduce more noise. To address this issue, we introduce Alignment Regularization (AReg) as a part of our framework. AReg is a model- and task-agnostic regularization term, which can easily be combined with existing embedding-based GSC models in a plug-and-play manner. The design of AReg is motivated by the analysis of GED in Section 3. Recall that GED is the difference between two graphs when they are best aligned under the optimal permutation $\pi^\star$, whose necessary conditions are that $\gamma_i$, $\gamma_j$ as well as $\|\gamma_i - \gamma_j\|_2$ take the minimum values. Thus, we expect that the learning-based model can couple with the nature of GED as much as possible, i.e., the

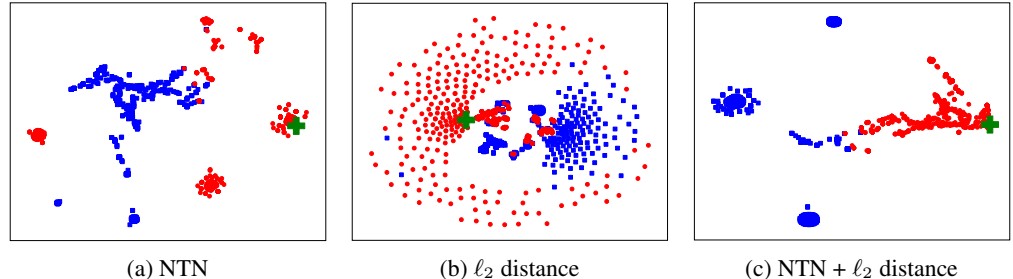

|            |                  |                       |
|------------|------------------|-----------------------|
| (a) NTN    | (b) $\ell_2$ distance | (c) NTN + $\ell_2$ distance |

Figure 4: t-SNE [29] visualization of the *IMDB* dataset. Each point is a graph encoded by the GNN encoder of ERIC. The green cross means a randomly sampled query graph; red points mean the top 50% of graph datasets that are most similar to the query graph based on ground-truth GEDs; blue points mean the remaining 50% graphs in the dataset. (a): Using NTN as the discriminator, many similar graphs do not cluster together around the query graph even if each cluster is tight. (b): Using $\ell_2$ distance as the discriminator, different clusters are separated clearly but the query graph is close to the cluster boundary; in addition each cluster is dispersive. (c): By adaptively combining NTN and $\ell_2$ distance, our model makes similar graphs closely located around the query, while dissimilar graphs are far away from the query.

GNN encoder can preserve the best alignment, in which case the difference between the learned graph representations can reflect the GED. To that end, we design the model based on the necessary conditions under $\pi^\star$. Specifically, based on Eq. (3) and Eq. (4), we instantiate $f_\theta(\cdot)$ as the $L$-layer Graph Isomorphism Network (GIN) [32], then at the $\ell$-th layer $f_\theta^{(\ell)}(\boldsymbol{A}_i[k])$ can be represented as:

$$\boldsymbol{H}_i^{(\ell)}[k] = f_\theta^{(\ell)}(\boldsymbol{A}_i[k]) = \text{MLP}_\theta^{(\ell)}\left((1 + \xi^{(\ell)})\boldsymbol{H}_i^{(\ell-1)}[k] + \boldsymbol{A}_i[k]\boldsymbol{H}_i^{(\ell-1)}\right), \tag{5}$$

where $\xi^{(\ell)}$ is a learnable parameter, $\boldsymbol{H}_i^{(\ell)} \in \mathbb{R}^{N \times d^{(\ell)}}$ is the feature matrix of $G_i$ at the $\ell$-th layer where $d^{(\ell)}$ denotes the feature dimension, $\boldsymbol{H}_i^{(0)} = \boldsymbol{X}_i$. On the other hand, AReg implements the readout function $g_\phi$ with a one-layer *permutation-invariant* DeepSets [33] to guarantee injectiveness, taking the form:

$$\boldsymbol{Z}_i^{(\ell)} = g_\phi^{(\ell)}(\boldsymbol{A}_i) = \text{MLP}_\phi^{(\ell)}\left(\sum_k^N f_\theta^{(\ell)}(\boldsymbol{A}_i[k])\right), \tag{6}$$

where $\text{MLP}_\phi^{(\ell)}$ and $\text{MLP}_\theta^{(\ell)}$ have the same output dimension $d_{\text{MLP}}^{(\ell)}$. Since $g_\phi(\boldsymbol{A}_j) = g_\phi(\pi^\star(\boldsymbol{A}_j))$, and $\gamma_i$ and $\gamma_j$ are sums over all $N$ nodes, hence they are permutation-invariant and we can remove all operation $\pi(\cdot)$ in Eq. (3) and Eq. (4). $\text{DIST}(\cdot, \cdot)$ can be defined as any distance metrics such as cosine similarity. Let $\gamma_i^{(\ell)}$ and $\gamma_j^{(\ell)}$ be the value of Eq. (3) and Eq. (4) at the $\ell$-th layer, by considering multi-scale cross-graph interactions, AReg is represented as $\mathcal{L}_{\text{AReg}}$ where:

$$\mathcal{L}_{\text{AReg}} = \frac{1}{L}\sum_\ell^L\left(\gamma_i^{(\ell)} + \gamma_j^{(\ell)} + \left\|\gamma_i^{(\ell)} - \gamma_j^{(\ell)}\right\|_2\right). \tag{7}$$

Since $\gamma_i$ and $\gamma_j$ induced from Eq. (2) integrate graph-level alignment and finer-grained node-level alignment, $\mathcal{L}_{\text{AReg}}$ therefore preserves underlying cross-graph interactions without computing complicated node-to-node similarity, also making the training stage more efficient.

### 4.2 Multi-Scale GED Discriminator

Now we have $L$ graph-level representations for $G_i$ and $G_j$ respectively, denoted as $\{\boldsymbol{Z}_i^{(1)}, \cdots, \boldsymbol{Z}_i^{(L)}\}$ and $\{\boldsymbol{Z}_j^{(1)}, \cdots, \boldsymbol{Z}_j^{(L)}\}$. We concatenate the layer-wise graph representations for $G$ as: $\widehat{\boldsymbol{Z}} = \bigoplus\{\boldsymbol{Z}^{(\ell)}\}_{\ell=1}^L$, where $\bigoplus$ denotes the concatenation operator along the last dimension to combine the graph representation in each layer, i.e., $\widehat{\boldsymbol{Z}}_i, \widehat{\boldsymbol{Z}}_j \in \mathbb{R}^{d_{\text{ms}}}$, and $d_{\text{ms}} = \sum_\ell d_{\text{MLP}}^{(\ell)}$. Then they are fed into a GED discriminator that generates score vectors as GED similarity embedding for the graph pair. Neural Tensor Network (NTN) [26] has demonstrated strong power to model the relation between the

graph-level embeddings of two graphs [1, 22, 30] thanks to its capacity of exploring the element-wise dependence among the features. However, directly using NTN as the discriminator may be expensive when the dimension of $\widehat{\boldsymbol{Z}}_i$ and $\widehat{\boldsymbol{Z}}_j$ is high. Thus, we decompose the weight matrix $\mathbf{W}^t \in \mathbb{R}^{d_{\mathrm{ms}} \times d_{\mathrm{ms}}}$ into two matrices $\mathbf{W}_1^t \in \mathbb{R}^{d_{\mathrm{ms}} \times d'}$ and $\mathbf{W}_2^t \in \mathbb{R}^{d' \times d_{\mathrm{ms}}}$ where $d' \ll d_{\mathrm{ms}}$ to reduce the number of parameters. Then NTN with decomposed weight matrices is used to measure the similarity between $\widehat{\boldsymbol{Z}}_i$ and $\widehat{\boldsymbol{Z}}_j$:

$$s_{\mathrm{NTN}}(G_i, G_j) = \delta_{\mathrm{NTN}} \left( \left[ (\widehat{\boldsymbol{Z}}_i \mathbf{W}_1^t)(\mathbf{W}_2^t \widehat{\boldsymbol{Z}}_j^\top) : t \in \{1, \cdots, T\} \right]^\top + \mathbf{W}_3 \bigoplus \left\{ \widehat{\boldsymbol{Z}}_i, \widehat{\boldsymbol{Z}}_j \right\} + \mathbf{b} \right), \tag{8}$$

where $\mathbf{W}_1 \in \mathbb{R}^{d_{\mathrm{ms}} \times d' \times T}$, $\mathbf{W}_2 \in \mathbb{R}^{d' \times d_{\mathrm{ms}} \times T}$, $\mathbf{W}_3 \in \mathbb{R}^{T \times 2d_{\mathrm{ms}}}$, and $\mathbf{b} \in \mathbb{R}^T$ are learnable; $T$ is a hyper-parameter controlling the output dimension, which is assigned as 16 for all datasets in our settings. $[\cdot]$ in Eq. (8) means computing $(\widehat{\boldsymbol{Z}}_i \mathbf{W}_1^t)(\mathbf{W}_2^t \widehat{\boldsymbol{Z}}_j^\top)$ for all $t \in \{1, \cdots, T\}$ and stacking them as a $T$-dimensional tensor. $\delta_{\mathrm{NTN}} : \mathbb{R}^T \to \mathbb{R}^{(0,1]}$ is a fully-connected neural network with Sigmoid activation as a regression function to project the similarity score vector to the final predicted similarity value. NTN is an expressive and general similarity discriminator because it can approximate many similarity measures. The first term of Eq. (8) can be regarded as a *multi-head* weighted cosine similarity function. It can also approximate kernel similarity between $\widehat{\boldsymbol{Z}}_i$ and $\widehat{\boldsymbol{Z}}_j$ according to the universal approximation theorem [13]. The second term captures the residual knowledge. However, it is difficult for NTN to approximate high-order Minkowski distance between $\widehat{\boldsymbol{Z}}_i$ and $\widehat{\boldsymbol{Z}}_j$, while diverse similarity discriminators may provide complementary information to reflect GED more accurately as shown in Fig. 4. Hence, we consider an additional similarity discriminator based on exponential $p$-order Minkowski distance:

$$s_p(G_i, G_j) = \delta_p \left( \exp \left( - \left\| \widehat{\boldsymbol{Z}}_i - \widehat{\boldsymbol{Z}}_j \right\|_p \right) \right), \tag{9}$$

where $\delta_p : \mathbb{R}^{d_{\mathrm{ms}}} \to \mathbb{R}^{(0,1]}$ is also a fully-connected neural network with Sigmoid activation. For simplicity, we uniformly set $p = 2$ (i.e., $\ell_2$ distance) for all datasets, and analyze the sensitivity of the hyper-parameter $p$ in Section 5.4. After two similarity scores $s_{\mathrm{NTN}}(G_i, G_j)$ and $s_p(G_i, G_j)$ are obtained, the final estimated similarity score is given by:

$$s(G_i, G_j) = \alpha s_{\mathrm{NTN}}(G_i, G_j) + \beta s_p(G_i, G_j), \tag{10}$$

where $\alpha$ and $\beta$ are trainable scalars denoting the weights of two similarity discriminators respectively. Given a graph database $\mathcal{D}$, the GED discriminator is trained on a set of $n$ training pairs $(G_i, G_j) \in \mathcal{D} \times \mathcal{D}$. The predicted similarity is compared against the ground-truth similarity $\mathbf{S}_{ij}$ based on GEDs with Mean Squared Error (MSE) loss function as:

$$\mathcal{L}_{\mathrm{GED}} = \frac{1}{n} \sum_{(G_i, G_j) \in \mathcal{D} \times \mathcal{D}} \mathrm{MSE}\left( s(G_i, G_j), \mathbf{S}_{ij} \right). \tag{11}$$

**Model training:**  Combining AReg and GED discriminator, the training stage aims to minimize the following overall objective function $\mathcal{L} = \mathcal{L}_{\mathrm{GED}} + \lambda \mathcal{L}_{\mathrm{AReg}}$, where $\lambda$ is an adjustable hyper-parameter controlling the strength of the regularization term.

**Model inference:**  Given a set of query graphs $\mathcal{Q}$ and a graph database $\mathcal{D}$, in the testing stage, all pairs of graphs $(G_i, G_j) \in \mathcal{Q} \times \mathcal{D}$ are fed into ERIC, and directly computing a similarity score for each pair based on Eq. (10) without any node-level interactions.

**Complexity:**  The time complexity of GIN in AReg is $O(m)$ [31] where $m$ is the number of edges. The cross-graph node-graph interactions have complexity $O(\max(N_i, N_j))$. The time complexity of the GED discriminator is $O(d_{\mathrm{ms}} d' T)$. Thus the complexity of ERIC is $O(m + \max(N_i, N_j) + d_{\mathrm{ms}} d' T)$ in the training stage, while in the inference stage the complexity is $O(m + d_{\mathrm{ms}} d' T)$.

## 5  Experiments

In this section, we empirically evaluate ERIC on the graph similarity computation task.

Table 1: Evaluation on benchmarks. **Bold** : best.

| | AIDS700 | | | | | LINUX | | | | | IMDB | | | | | NCI109 | | | | |
|---|---|---|---|---|---|---|---|---|---|---|---|---|---|---|---|---|---|---|---|---|
| | mse (×10⁻³)↓ | ρ↑ | τ↑ | p@10↑ | p@20↑ | mse (×10⁻³)↓ | ρ↑ | τ↑ | p@10↑ | p@20↑ | mse (×10⁻³)↓ | ρ↑ | τ↑ | p@10↑ | p@20↑ | mse (×10⁻³)↓ | ρ↑ | τ↑ | p@10↑ | p@20↑ |
| Beam | 12.090 | 0.609 | 0.463 | 0.481 | 0.493 | 9.268 | 0.827 | 0.714 | 0.973 | 0.924 | - | - | - | - | - | - | - | - | - | - |
| VJ | 29.157 | 0.517 | 0.383 | 0.310 | 0.345 | 63.86 | 0.581 | 0.450 | 0.287 | 0.251 | - | - | - | - | - | - | - | - | - | - |
| Hungarian | 25.296 | 0.510 | 0.378 | 0.360 | 0.392 | 29.81 | 0.638 | 0.517 | 0.913 | 0.836 | - | - | - | - | - | - | - | - | - | - |
| SimGNN | 1.573 | 0.835 | 0.678 | 0.417 | 0.489 | 2.479 | 0.912 | 0.791 | 0.635 | 0.650 | 1.437 | 0.871 | 0.752 | 0.710 | 0.769 | 7.767 | 0.576 | 0.435 | 0.023 | 0.040 |
| GraphSim | 2.014 | 0.839 | 0.662 | 0.401 | 0.499 | 0.762 | 0.953 | 0.882 | 0.956 | 0.951 | 1.924 | 0.825 | 0.821 | 0.813 | 0.825 | 8.752 | 0.557 | 0.497 | 0.086 | 0.032 |
| GMN | 4.610 | 0.672 | 0.497 | 0.200 | 0.263 | 2.571 | 0.906 | 0.763 | 0.888 | 0.856 | 4.320 | 0.665 | 0.601 | 0.588 | 0.593 | 11.710 | 0.336 | 0.358 | 0.017 | 0.019 |
| EGSC | 1.676 | 0.888 | 0.723 | 0.604 | 0.708 | 0.214 | 0.984 | 0.897 | 0.987 | 0.989 | 0.573 | **0.939** | **0.829** | 0.872 | 0.883 | 9.356 | 0.545 | 0.414 | 0.055 | 0.078 |
| MGMN | 2.297 | 0.904 | 0.736 | 0.456 | 0.534 | 2.040 | 0.965 | 0.858 | 0.956 | 0.920 | 0.496 | 0.881 | 0.803 | 0.874 | 0.861 | 9.631 | 0.492 | 0.426 | 0.015 | 0.051 |
| ERIC | **1.383** | **0.906** | **0.740** | **0.679** | **0.746** | **0.113** | **0.988** | **0.908** | **0.994** | **0.996** | **0.385** | 0.890 | 0.791 | **0.882** | **0.891** | **7.127** | **0.591** | **0.525** | **0.118** | **0.080** |

## 5.1 Datasets

We conduct experiments on four widely used GSC datasets including *AIDS700*, *LINUX*, *IMDB* [1], and *NCI109* [2]. Following the same splits as [1–3], i.e., 60%, 20%, and 20% of all graphs as training set, validation set, and query set, respectively. The training set together with the validation set are called the *database*. More details of the datasets are presented in Appendix B.1.

## 5.2 Experimental Settings

**Baselines.** We implement two groups of baselines for comparison: (1) Combinatorial search-based Methods: Beam [20], Hungarian [15], and VJ [9]; (2) GNN-based Methods: SimGNN [1], GMN [16], GraphSim [3], MGMN [18], and EGSC [22]. We re-implemented all baselines with the same hyper-parameters as the original literature provides, or carefully tuned the parameters to get the optimal results when they are not provided. We give more implementation details of ERIC and baselines in Appendix B.2.

**Evaluation Metric.** To comprehensively evaluate our model on the similarity computation task, following [1, 22] five metrics are adopted to evaluate results for fair comparisons: **Mean Squared Error** (MSE) which measures the average squared differences between the predicted and the ground-truth similarity. **Spearman's Rank Correlation Coefficient** ($\rho$) and **Kendall's Rank Correlation Coefficient** ($\tau$) which evaluates the ranking correlations between the predicted and the true ranking results. **Precision@$k$** where $k = 10, 20$, which computes the interactions of the predicted and ground-truth top-$k$ results divided by $k$. The smaller the MSE, the better performance of models; for $\rho$, $\tau$, and $p@k$, the larger the better.

## 5.3 Main Results

The results of ERIC and the baselines on our benchmarks are reported in Table 1. For datasets with relatively small graphs whose ground-truth GEDs are exactly computed by the $A^\star$ algorithm, ERIC consistently achieves state-of-the-art performance across all evaluation metrics as shown in Table 1. For datasets with large graphs whose ground-truth GEDs are computed approximately, ERIC still achieves the best results on NCI109, and comparable results on IMDB. The suboptimal performance of the methods based on node-to-node similarity (SimGNN, GraphSim, GMN, and MGMN) demonstrate that overly dense and fine-grained similarity computation may not always bring a beneficial boost. EGSC also uses GIN as the backbone and totally ignores cross-graph node-level interactions, and it achieves the best results on metrics $\rho$ and $\tau$ on the IMDB dataset. The reason is that EGSC adopts intra-graph attention pooling and NTN in all layers to enhance the expression ability of representation vectors. However, the attention mechanism and full NTN need additional parameters which increase the burden of learning. Although EGSC proposed a student model based on knowledge distillation to speed up the inference process, it would sacrifice the prediction accuracy in most cases. The performance of ERIC over baselines illustrates the importance of fine-grained interactions and the proper design of such interactions.

## 5.4 Ablation Study

**Effectiveness of AReg and Multiple GED Discriminators.** ERIC contains two key components: AReg and Multi-Scale GED Discriminator. To glean a deeper insight into how different components help ERIC to achieve highly competitive results, we conduct ablation experiments by removing individual components separately. Specifically, we alter the loss by removing the AReg term to

Table 2: Ablation study on the key components of ERIC on AIDS700 and LINUX.

| | AIDS700 | | | LINUX | | |
|---|---|---|---|---|---|---|
| | mse | $\rho$ | $p@10$ | mse | $\rho$ | $p@10$ |
| ERIC | **1.383** | **0.906** | **0.679** | **0.113** | **0.988** | **0.994** |
| ERIC (w/o AReg) | 1.573 | 0.886 | 0.652 | 0.363 | 0.965 | 0.979 |
| ERIC (w/o NTN) | 1.687 | 0.854 | 0.633 | 0.302 | 0.951 | 0.969 |
| ERIC (w/o $\ell_2$) | 1.466 | 0.881 | 0.674 | 0.253 | 0.974 | 0.980 |

Table 3: Transferability study of AReg on AIDS700 and LINUX.

| | AIDS700 | | | LINUX | | |
|---|---|---|---|---|---|---|
| | mse | $\rho$ | $p@10$ | mse | $\rho$ | $p@10$ |
| SimGNN | 1.573 | 0.835 | 0.417 | 2.479 | 0.912 | 0.635 |
| SimGNN+AReg | **1.439** | **0.858** | **0.506** | **1.974** | **0.945** | **0.658** |
| EGSC | 1.676 | 0.888 | 0.604 | 0.214 | 0.984 | 0.987 |
| EGSC+AReg | **1.478** | **0.904** | **0.643** | **0.142** | **0.989** | **0.992** |

Table 4: Inference time (sec).

| Dataset | SimGNN | GraphSim | GMN | MGMN | EGSC | ERIC |
|---|---|---|---|---|---|---|
| AIDS700 | 10.773 | 14.043 | 23.975 | 11.337 | 8.763 | **6.662** |
| LINUX | 19.347 | 31.238 | 82.489 | 22.574 | 21.573 | **18.969** |
| IMDB | 225.682 | 379.480 | 1253.551 | 357.933 | 133.437 | **48.750** |
| NCI109 | 2913.178 | 3463.620 | $> 10^4$ | 3726.834 | 2097.405 | **1763.356** |

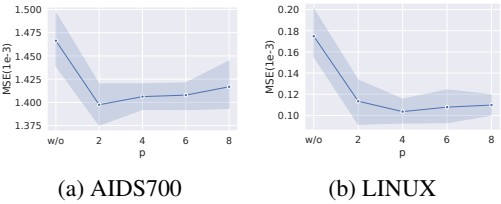

(a) AIDS700    (b) LINUX

Figure 5: Impact of different order $p$ on AIDS700 and LINUX datasets.

study the effect of the cross-graph node-graph interaction, with the results reported in ERIC (w/o AReg) of Table 2. We find that only using the MSE loss $\mathcal{L}_{\mathrm{GED}}$ will lead to a performance drop on the extracted three evaluation metrics, which confirms the necessity of AReg. In our design, the final similarity score is obtained by the weighted average of two similarity scores based on the NTN discriminator $s_{\mathrm{NTN}}$ and the $\ell_2$ discriminator $s_p$ respectively. To further investigate the impact of multiple GED discriminators, we remove one of $s_{\mathrm{NTN}}$ and $s_p$ to study the effect of each discriminator. As shown in Table 2, both ERIC (w/o NTN) and ERIC (w/o $\ell_2$) cause a decrease in effectiveness, which demonstrates both of them contribute to the final performance, while the model benefits more from NTN.

**Transferability of AReg** Further, since AReg is a model-agnostic regularization term, we are interested in the transferability of AReg, so we evaluate the performance of applying AReg to other GSC models. We use SimGNN and EGSC as baselines. For SimGNN, we use AReg to replace the node-to-node similarity computation. For EGSC, we directly add AReg to the loss function. Table 3 shows the effect of AReg on SimGNN and EGSC. As expected, the advantage of SimGNN+AReg over SimGNN shows that integrating the dense similarity matrix into the final similarity score may bring noise which affects performance. While the performance on EGSC proves that combining fine-grained similarity in a proper way, i.e., node-graph rather than node-node, can improve the model.

**Sensitivity of Order $p$ in $\delta_p$** In the multi-scale GED discriminator module, we adopt the exponential $p$-order Minkowski distance as a similarity measure to further improve the separability of clusters in the graph embedding space. For simplicity, we directly use the 2-order Minkowski distance, i.e., $\ell_2$ distance. Results in Table 2 show the effectiveness of considering the $\ell_2$ distance, and Fig. 4 demonstrates the complementarity of different similarity measures. To investigate the sensitivity of $p$ in our model, we set $p$ from $[2, 4, 6, 8]$, and run 10 times for each value of $p$. Then the results of mean square error with standard deviation are reported in Fig. 5. We can observe that the performance increases first and then the error becomes stable when $p \geq 2$. It shows that using the $\ell_2$ distance as the GED discriminator is suitable for our model and higher-order Minkowski distance would not improve the performance.

## 5.5 Inference Time

In ERIC, the cross-graph alignment only acts as a regularization term in the training stage but is no longer used in the inference stage. To evaluate the efficiency, we compare the performance of ERIC with baselines in terms of inference time in Table 4. In Table 5 of Appendix B.1 we list the number of graph pairs in the inference stage (#Testing pairs), and all experiments are implemented with a single machine with 1 NVIDIA Quadro RTX 8000 GPU. As can be observed, node-to-node

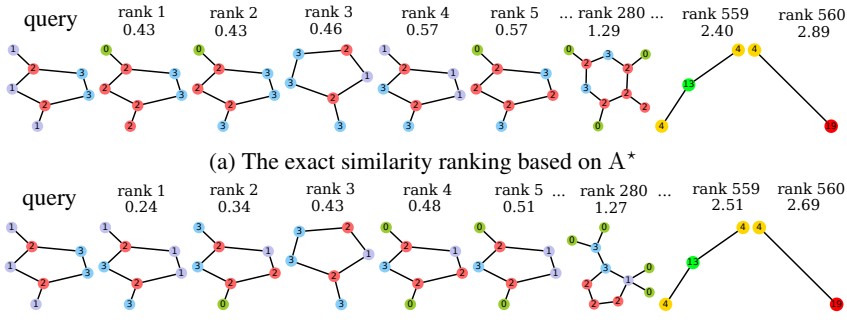

(a) The exact similarity ranking based on A⋆

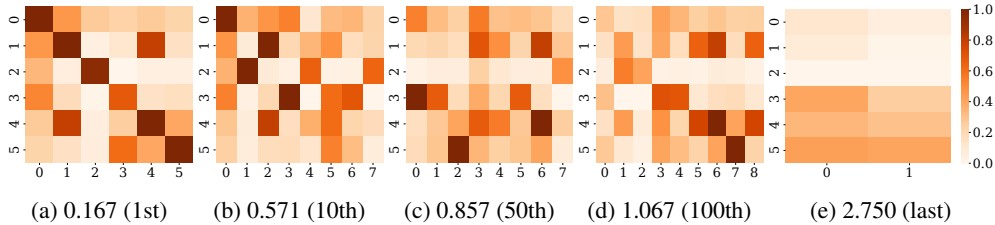

(b) The predicted similarity ranking based on ERIC

Figure 6: Visualization of graph search examples on the AIDS700 dataset. Nodes with different labels are assigned different colors. (a) and (b) are similarity rankings based on the normalized GEDs computed by A⋆ (exact) and ERIC (estimation) respectively.

| (a) 0.167 (1st) | (b) 0.571 (10th) | (c) 0.857 (50th) | (d) 1.067 (100th) | (e) 2.750 (last) |

Figure 7: Heatmaps of cross-graph node-to-node cosine similarity based on the node representations learned by the GNN encoder of ERIC. Y-axis means the node in a randomly sampled query graph. (a)~(e) are ranked by the exact normalized GEDs between the query graph and graphs of particular ranks in the database. For example, 0.571 (10th) means the nGED of the graph that ranks 10th in similarity to the query graph is 0.571. The color depth in the heatmap represents the similarity of the node pair; the deeper the color, the higher the similarity.

similarity computation is more time-consuming on all datasets. Our proposed node-graph similarity computation does not incur substantial additional running time in practice. To summarize, ERIC is faster than all baseline models, while still achieving significantly higher accuracy on all datasets.

## 5.6 Visualization of Graph Search

The goal of the graph search task is to find $k$ graphs from the dataset that are most similar to the given query graph, which is a routine task in drug discovery [24]. In Fig. 6 we show a case based on the AIDS700 dataset, where each graph represents a functional group. Comparing the exact similarity ranking computed by A⋆ and the estimated one computed by ERIC, we see that the ranking of ERIC has a high consistency with the exact ranking, which proves that ERIC is able to extract graphs that contain the similar substructure from around 700 graphs with varying size and structure and the top-ranked graph has a high degree of isomorphism with the query. It also demonstrates that ERIC can capture structural patterns shared across graphs. More results of visualization are provided in Appendix C.

## 5.7 Analysis of Node-to-Node Similarity

In Fig. 7, we show the node-to-node similarity between a query graph and the graphs at different ranking positions in terms of normalized GEDs. It can be found that for graph pairs with a small GED, the node representations generated by ERIC show a clear correspondence between nodes as shown in Fig. 7 (a). As the GED increases, the correspondence between nodes across the graph gradually weakens, i.e., cross-graph node-to-node similarity reduces. Thus, the similarity matrices in Fig. 7 can guide us to find a better alignment between pair-wise graphs.

# 6 Related Work

The goal of graph similarity computation (GSC) is to quantify the similarity between graphs under a specific similarity measure. Various similarity measures have been well studied in prior works, such as graph edit distance (GED) [5, 25, 20] and maximum common subgraph (MCS) [6, 10]. Among these, GED is the most popular one, and many other similarity measures can be proven to be its special cases [17]. However, computing the exact GED between two graphs is known to be NP-hard. In practice, the computation becomes challenging when the number of nodes is more than 16 [4] using exact GED solvers such as the $A^\star$ algorithm [25]. Thus, approximate algorithms have been proposed for GED-based GSC. These approximate methods can be broadly divided into two classes: (1) *Combinatorial search-based methods*, which aim to exploit combinatorial structures or theoretical lower-bounds to approximate GED. Beam [20] is a GED estimator based on Beam Search; Hungarian [15] is proposed based on the famous Hungarian algorithm for weighted graph matching; Hausdorff approximation [11] provides a lower bound for the GED approximation; VJ [9] uses the Volgenant and Jonker algorithm for GED approximation. However, these methods are highly heuristic-driven and run with either sub-exponential time or cubic time complexity, limiting the scalability as the graphs grow in size. Also, these methods totally ignore the node feature information, so that the underlying semantic similarity can not be captured. (2) *Learning-based methods*, which are data-driven and aim to learn graph similarity from the data itself, hopefully with higher accuracy and far lower time costs compared with search-based methods. GMN [16] introduces a cross-graph attention layer that allows the nodes in the two graphs to interact with each other and predicts graph similarity using the representation vectors that fuse cross-graph information. SimGNN [1] relies on a shared GNN encoder, a neural tensor network, and a pairwise node comparison module to compute the similarity between two graphs. GraphSim [3] extends SimGNN by using convolutional neural networks to capture the multi-scale node-level interactions. EGSC [22] simplifies SimGNN by ignoring node-level interactions and uses knowledge distillation to accelerate the inference stage. MGMN [18] employs a node-graph matching network to capture cross-level features between nodes of a graph and the other whole graph, where the cross-graph aggregation weights are computed by node-to-node attention coefficients. Our proposed ERIC is also a learning-based method. Unlike the above approaches that either discard node-level interactions entirely, or performs interactions between all node pairs, our method proposes a novel node-graph interaction paradigm that avoids dense similarity computation while preserving fine-grained interactions.

# 7 Conclusion

We propose ERIC, a simple yet powerful GNN-based framework for the graph similarity computation task. Specifically, we first give a deep insight into the graph edit distance, and propose Alignment Regularization (AReg) which is a separated structure independent of the end-to-end learning pipeline. AReg frees the model from complicated node-to-node interaction for similarity computation. Further, we propose a multi-scale GED discriminator to improve the discriminative ability of the learned representations. We show the effectiveness of our model through a comprehensive set of experiments and analyses.

## Acknowledgments and Disclosure of Funding

This work is supported in part by Shenzhen Baisc Research Fund under grant JCYJ20200109142217397.

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
