# A  Sufficiency Analysis of AReg

As mentioned in Section 3, GED measures the difference between two graphs when they are best aligned. In other words, if we perform node permutation on one graph, so that the same-indexed nodes in two graphs are as similar as possible in structure, then the GED can be directly computed by computing the difference between the permuted adjacency matrices. In the paper, we refer to the permutation that satisfies such a condition as the optimal permutation $\pi^\star$. However, directly traversing all possible node permutations to find $\pi^\star$ has complexity $O(N!)$. Alternatively, we rethink this problem in embedding space, and consider necessary conditions that are implicitly required by $\pi^\star$ and that are easy to implement in the model. To that end, we define $\gamma_i$, $\gamma_j$ and require both objectives, as well as their difference $||\gamma_i - \gamma_j||_2$ to be minimized. To analyze the accuracy of such approximation, we analyze the sufficiency of these conditions. Notice that $\gamma_i = \sum_k^N ||\mathrm{DIST}\left(f_\theta(\boldsymbol{A}_i[k]), g_\phi(\boldsymbol{A}_i)\right) - \mathrm{DIST}\left(f_\theta(\boldsymbol{A}_i[k]), g_\phi(\pi(\boldsymbol{A}_j))\right)||_2$ formally reflects how each node is compatible with the two graphs equally well. A minimum $\gamma_i$ therefore implies the two graphs are properly aligned. Similarly, a minimum $\gamma_j$ achieves the same goal from the standpoint of $G_j$. As an additional perspective, we view $2N$ vectors $\{f_\theta(\boldsymbol{A}_i[k])\}_{k=1}^N \cup \{f_\theta(\pi(\boldsymbol{A}_j)[k])\}_{k=1}^N$ as a set of anchors, denoted as $\mathcal{T}$. Since $f_\theta(\cdot) \in \mathbb{R}^d$, if the linear span of a subset of $\mathcal{T}$ can be a space $\mathbb{R}^d$, then for all $k$, a small difference between $\mathrm{DIST}\left(f_\theta(\boldsymbol{A}_i[k]), g_\phi(\boldsymbol{A}_i)\right)$ and $\mathrm{DIST}\left(f_\theta(\boldsymbol{A}_i[k]), g_\phi(\pi(\boldsymbol{A}_j))\right)$ can reflect a small distance between $g_\phi(\boldsymbol{A}_i)$ and $g_\phi(\pi(\boldsymbol{A}_j))$, thus we can further get that $\boldsymbol{A}_i$ and $\pi(\boldsymbol{A}_j)$ are very similar input for the injective function, i.e., $\boldsymbol{A}_i$ and $\pi(\boldsymbol{A}_j)$ is best aligned, so we have $\pi = \pi^\star$. If $||\gamma_i - \gamma_j||_2$ is minimized meanwhile, it will lead to the distance between $f_\theta(\boldsymbol{A}_i[k])$ and $f_\theta(\pi(\boldsymbol{A}_j)[k])$ takes the minimum, which is the node-level embeddings under $\pi^\star$. Thus, assuming $N > d$, as long as at least $d$ of the $2N$ nodes have different neighbor indexes, there must exist a subset of $\mathcal{T}$ that can span $\mathbb{R}^d$, then our necessary conditions can be sufficient. In most real-world cases the number of nodes $N$ is large than the embedding dimension $d$, and the connected structures of most nodes are not exactly the same due to the complexity of the network. Hence, the sufficiency of our proposed conditions usually can be satisfied in real-world cases.

# B  More Details of Dataset and Experiments

## B.1  Datasets

The statistical information of our experimental datasets is shown in Table 5. The exact GEDs of AIDS700 and LINUX are computed by the $\mathrm{A}^*$ [25] algorithm in exponential time. For datasets with large graphs, following [1, 2] IMDB and NCI109 take the pair-wise smallest distance computed by three approximate algorithms, Beam [20], Hungarian [15], and VJ [9], as ground truth.

Table 5: Dataset statistics.

| Dataset | #Graphs | AVG #Nodes | AVG #Edges | #Features | #Queries | #Testing Pairs |
|---|---|---|---|---|---|---|
| AIDS700 | 700 | 8.9 | 8.8 | 29 | 140 | 78,400 |
| LINUX | 1,000 | 7.6 | 6.9 | 1 | 200 | 160,000 |
| IMDB | 1,500 | 13.0 | 65.9 | 1 | 300 | 360,000 |
| NCI109 | 4,127 | 29.6 | 32.1 | 38 | 400 | 2,726,626 |

## B.2  Implementation Details

For the AReg submodule, the number of layers of the GNN encoder is set to 4 and the hidden dimension of each layer is selected from $[32, 64]$. The DeepSets function $\mathrm{MLP}_\phi$ is a 1-layer MLP with the same output dimension with the GNN encoder in each layer. We simply set $\mathrm{DIST}(\cdot, \cdot)$ to cosine similarity. For the GED discriminator submodule, both the hyper-parameter $d'$ and the output dimension $T$ of the decomposed NTN are set to 16. Both $\delta_{\mathrm{NTN}}$ and $\delta_p$ are 2-layer MLP with hidden dimension 16. For SimGNN and GraphSim, the results we re-implement using the official code are different from that reported by original papers. It is because SimGNN and GraphSim adopt an additional hyper-parameter to scale the ground-truth GEDs, while other baselines do not. Thus, we do not consider scaling of the ground-truth. The source code is available https://github.com/JhuoW/ERIC.

# C   Extra Visualizations of Graph Search

A few more graph search cases are included in Fig. 8 and Fig. 9, which show that the ranking results learned by our ERIC have high consistency with the ground-truth similarity ranking. We can find that graphs at the same ranking position have similar substructures. It indicates that ERIC is able to retrieve graphs that best resemble the query.

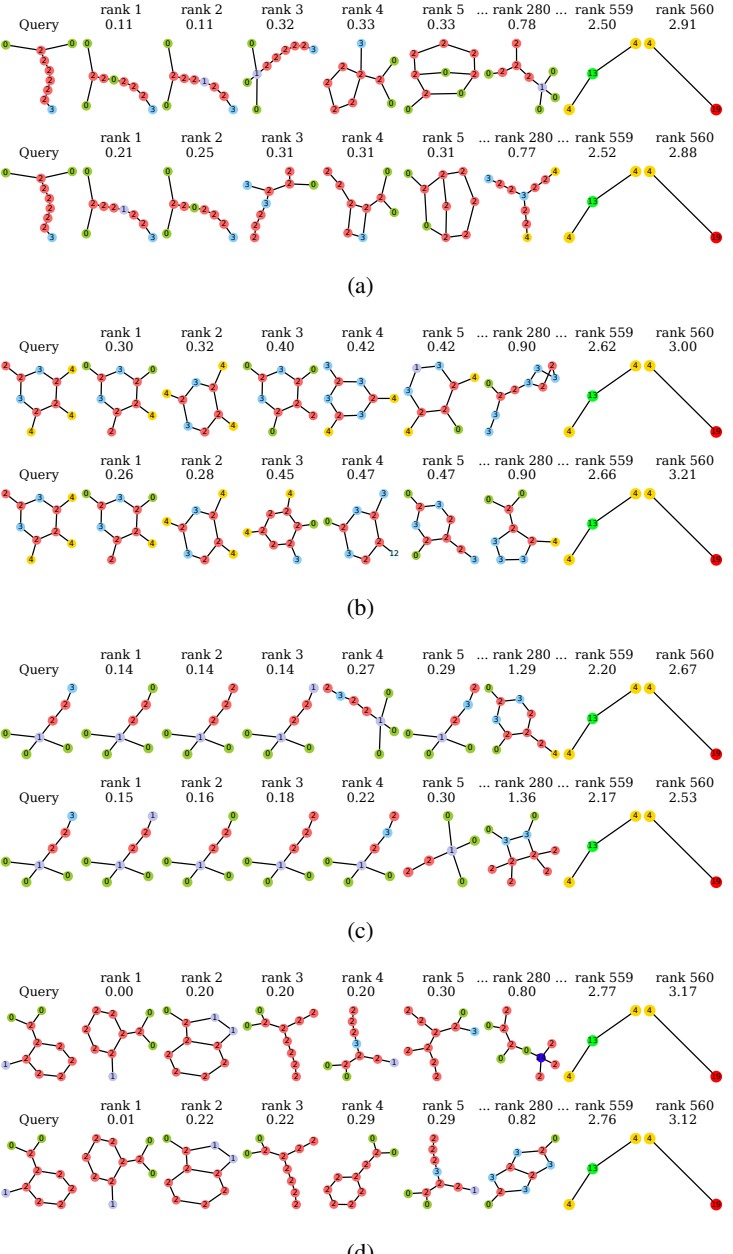

Figure 8: Visualization of graph search results on the AIDS700 dataset. Nodes with different labels are assigned different colors. The first line of each sub-figure is the graph search results ranked by $A^\star$, and the second line is ranked by ERIC.

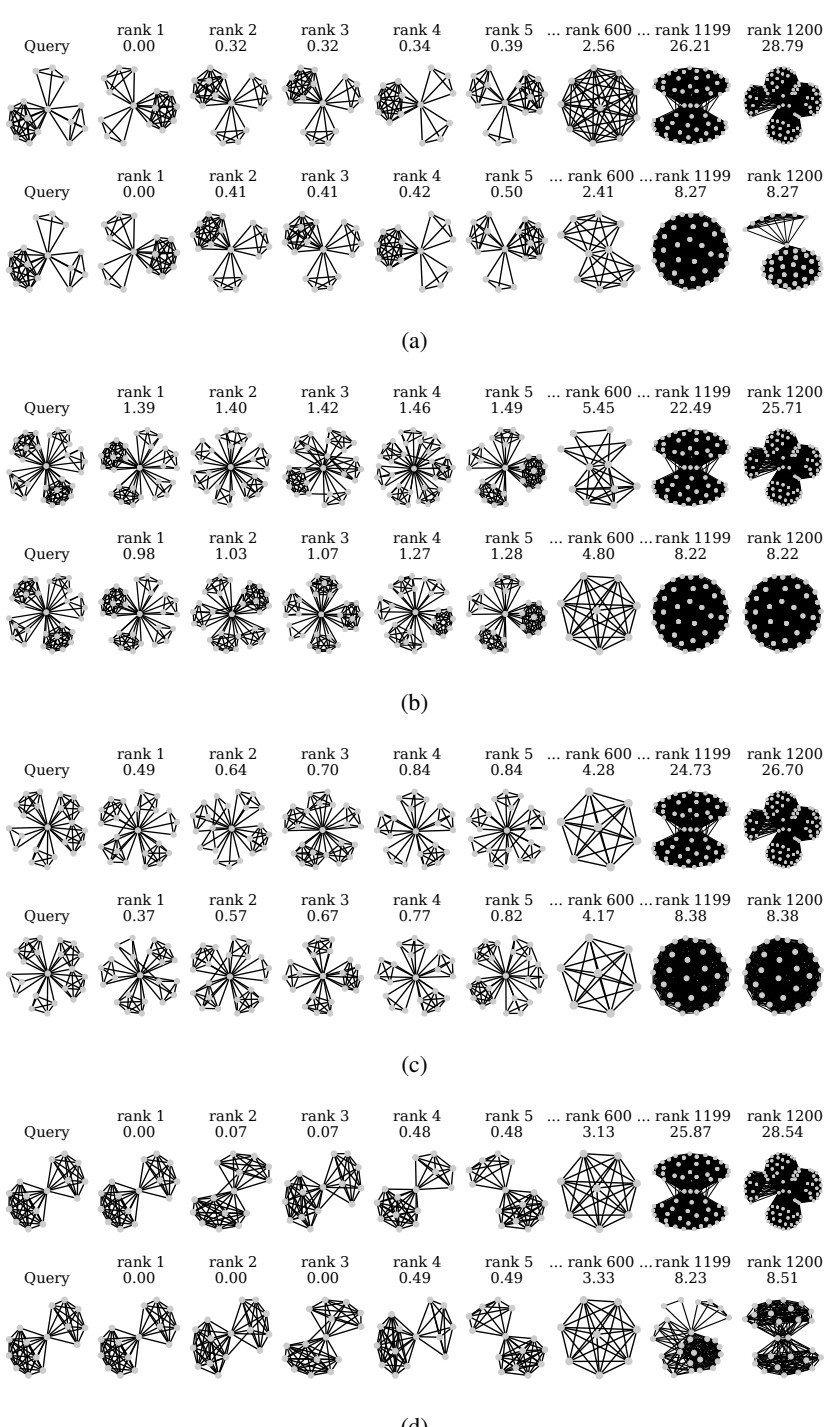

Figure 9: Visualization of graph search results on the IMDBMulti dataset. The first line of each sub-figure is the graph search results ranked by approximate algorithms, and the second line is ranked by ERIC.