# OpenReview forum: "Efficient Graph Similarity Computation with Alignment Regularization"
_NeurIPS.cc/2022/Conference — NeurIPS 2022 Accept_

### Official Review · Reviewer_z4ZB · 2022-07-05

**Rating:** 5
**Confidence:** 4
**Soundness:** 3 good
**Presentation:** 2 fair
**Contribution:** 3 good

**Summary:**

This paper studies the efficient graph similarity computation problem. The core is to compute graph edit distance, which is an NP-hard problem. Recent efforts resort to graph network for approximation. However, the existing node-level matching module is usually computationally expensive. This work shows that the node-level matching is not necessary. Instead, the authors propose a node-graph level correspondence as a regularization term, and inject it into the standard training framework. The experimental results show that the proposed regularization term performs competitively in both matching accuracy and inference speed.

**Questions:**

-Line 40, it is stated that “since cross-graph interactions are ignored, …”. While EGSC seems have the early fusion layers which aim to learn the cross-graph interaction. Please clarify this sentence.

-Line 209, $\lambda$ is a trainable parameter. Please elaborate on how it is trained. If $\lambda$ is prefixed, please provide the sensitivity analysis on it.

-Table 4 in supplementary, it seems that the performance gain by AReg is limited. For instance, in AIDS700, ERIC (w/o AReg) on mse/$\rho$/P@10 are already 1.573/0.886/0.652. And it already achieves the best performance in p@10 compared with other baselines. Why ERIC w/o AReg can beat other baselines like EGSC?

Also, ablation study is an important analysis, deserves to be put in the main text. I suggest the authors better organize the content to fit this part in the main paper.


**Limitations:**

The authors adequately addressed the limitations and potential negative social impact of their work.

**Strengths And Weaknesses:**

Strength

-This paper is mostly well written. I like the discussion in the motivation section, from which the proposed formulation is derived.

-The proposed regularization term AReg, capturing the node-graph interaction between two graphs, is intuitive.

-The experimental results suggest the proposed AReg outperforms other baselines in both accuracy and inference speed.


Weakness

-Some technical points are unclear in terms of presentation. (see questions)

-Ablated results suggested that the proposed AReg has limited contribution to the final performance.

-Missing key parameter analysis on $\lambda$. It is unclear to me how $\lambda$ is learned.

---

> ### Author Response · Authors · 2022-08-02
> **Respond to reviewer z4ZB**
>
> We appreciate the thorough reviews. We answer your questions below.
>
> **Q1**: Line 40, it is stated that “since cross-graph interactions are ignored, …”. While EGSC seems have the early fusion layers which aim to learn the cross-graph interaction. Please clarify this sentence.
>
> **A1**: In its early-fusion layer, EGSC first generates _graph-level representations_ for two input graphs respectively, and then concatenates them to generate a summarized representation vector, and feeds it into a neural network to output a similarity score for a pair of input graphs. Indeed, EGSC learns cross-graph interactions, but in a coarse-grained way, because the interactions are at _graph-level_, ignoring the **node-level** interactions, which contain finer-grained similarity information as we mentioned in Line 41. Our method implicitly considers node-level interactions with an efficient regularization technique, AReg. Thank you for pointing out our inaccurate statement, we will re-state this problem of EGSC in the updated version of the paper.
>
>
>
> **Q2:** Line 209, $\lambda$ is a trainable parameter. Please elaborate on how it is trained. If $\lambda$ is prefixed, please provide the sensitivity analysis on it.
>
> **A2：** In our work, the weight $\lambda$ of AReg term in the loss function is an adjustable parameter that can be preset before the training stage, or be searched by some hyper-parameter search algorithm. Thus,  'trainable' used here is inaccurate and we have modified it to 'adjustable' in the updated version. To analyze the sensitivity of $\lambda$, we set its value from 0 to 2, and conduct experiments with different $\lambda$ on AIDS700 and LINUX datasets, and report the MSE results as follows:
>
> | $\lambda$ | 0     | 0.2   | 0.4   | 0.6   | 0.8       | 1.0       | 1.2   | 1.4   | 1.6   | 1.8   | 2.0   |
> | --------- | ----- | ----- | ----- | ----- | --------- | --------- | ----- | ----- | ----- | ----- | ----- |
> | AIDS700   | 1.573 | 1.552 | 1.467 | 1.414 | **1.374** | 1.391     | 1.388 | 1.405 | 1.593 | 1.628 | 1.776 |
> | LINUX     | 0.363 | 0.307 | 0.315 | 0.224 | 0.135     | **0.107** | 0.118 | 0.137 | 0.259 | 0.321 | 0.317 |
>
> We can observe that as $\lambda$ increases from zero, the performance increases until reaches at a peak, and then decreases. This is reasonable as the optimization objective of ERIC is to minimize $\mathcal{L} = \mathcal{L}\_{GED} + \lambda \mathcal{L}\_{AReg}$, where the Alignment Regularization term $\mathcal{L}\_{AReg}$ is used to guide graph alignment, and $\mathcal{L}\_{GED}$ helps to learn the distance between the representations. If $\lambda$ is too small, the alignment operation is weakened, thus the distance between them cannot reflect GED accurately. On the other hand, if $\lambda$ is too large, the multi-scale discriminator may not be fully trained to fit the ground-truth GED. We can also find the best value of $\lambda$ is around 1, which indicates that the alignment regularization and GED discriminator are both important for learning similarity score.
>
> **Q3:** Table 4 in supplementary, it seems that the performance gain by AReg is limited. For instance, in AIDS700, ERIC (w/o AReg) on mse/ρ/P@10 are already 1.573/0.886/0.652. And it already achieves the best performance in p@10 compared with other baselines. Why ERIC w/o AReg can beat other baselines like EGSC?
>
> **A3:** Indeed, on the AIDS700 dataset, our ERIC (w/o AReg) already achieves the best performance. Comparing the state-of-the-art baselines EGSC and SimGNN which use NTN as a single GED discriminator, the main difference of ERIC(w/o AReg) is that we adopt multi-scale GED discriminator as described in Sec.4.2.  Specifically, we not only use NTN as GED discriminator, but also adopt $\ell_2$-distance as an additional discriminator which directly computes the distance between the representations of pair-wise graphs without introducing any parameters. Such a simple design brings consistent benefits. Fig.4 shows  $\ell_2$ distance and NTN  play complementary roles, which is the main reason why ERIC without AReg can still outperform EGSC and SimGNN.  Besides, although ERIC(w/o AReg) already performs well, our proposed AReg still has an important impact on the model. For instance, for AIDS700, ERIC(w/o AReg) achieves 1.573/0.886/0.652 for MSE, $\rho$ and p@10 respectively, and ERIC achieves 1.374/0.906/0.685, which are 12.7%, 2.2%, 5.1% relative improvements over ERIC(w/o AReg). For LINUX dataset, ERIC(w/o AReg) achieves 0.363/0.965/0.979 for MSE, $\rho$ and p@10, while ERIC achieves 0.107/0.988/0.994, which are 70.5%, 2.4% and 1.5% relative improvements. Taking a panoramic view of all the experimental results, it is not a marginal improvement. Thus, the ablation study demonstrates the effectiveness of AReg term.
>
> **Q4:** Ablation study is an important analysis, deserves to be put in the main text.
>
> **A4:** Thanks for your valuable suggestion. We will follow your suggestion and re-organize our paper.

---

> > ### Comment · Reviewer_z4ZB · 2022-08-08
> > **Thanks for the response**
> >
> > Thank you for the response. It primarily addresses my questions. The inaccurate statement on $\lambda$ is trainable does raise a flag to me, but it is good to see the authors admit and correct it and provide the parameter analysis as requested. So I'll not drop my rating, and keep it borderline accepted.
> >
> > Please make sure you incorporate these new changes and discussions into the final version.

---

> ### Author Response · Authors · 2022-08-07
> **Thank you for the time and we hope our response helps to address your questions.**
>
> Dear Reviewer z4ZB,
>
> We sincerely hope our posted response can help to address your concern. If you have any further comments and questions, please let us know and we are glad to write a follow-up response.
>
> Best regards,
>
> The Authors

---

### Official Review · Reviewer_HgEJ · 2022-07-10

**Rating:** 6
**Confidence:** 3
**Soundness:** 2 fair
**Presentation:** 3 good
**Contribution:** 3 good

**Summary:**

This paper proposes a novel framework, named ERIC, for graph similarity computation. The first technical contribution is to analyze GED in embedding space and further approximate the ground-truth GEDs with $\gamma_{i}$, $\gamma_{j}$ and $\||\gamma_{i}-\gamma_{j}\||_{2}$. Based on the analysis, the authors introduce Alignment Regularization (AReg) into model training. In addition, the authors discuss the necessity of multi-scale GED discriminator when training GNN encoder. Overall, the graph similarity computation framework ERIC consists of AReg and Multi-Scale GED Discriminator as two submodules.

**Questions:**

As mentioned in **Weaknesses**, I suggest the authors to:
1. Take more types of discriminators into account;
2. Analyze the accuracy of this approximation method;

**Limitations:**

I think the applicability and practicality of the proposed method is not low, but the authors describe little about its social value and impact. Nor do they mention the limitations of the paper. It would be better to describe more about social impacts or limitations.

**Strengths And Weaknesses:**

**Strengths**:
1. This paper is well structured, clearly written and easy-to-follow. In section 3, the authors first theoretically investigate into GED in embedding space. The corresponding conclusion supports the justification of AReg in section 4.
2. Approximating the ground-truth GEDs with $\gamma_{i}$, $\gamma_{j}$ and $\||\gamma_{i}-\gamma_{j}\||_{2}$ is novel and reasonable. This method not only reduces the computational and memory cost, but also simplifies the matching process and thus makes it easy to implement.
3. Figure 4 provides an intuition that multiple similarity discriminators are likely to provide more useful information to reflect GED. It is a good practice.
4. The authors conduct extensive experiments and illustrate the results clearly in the paper and the supplementary material.

**Weaknesses**:
1. The authors claim that *diverse similarity discriminators may provide complementary information to reflect GED more accurately*. However, they only discuss and use two kinds of similarity discriminators, i.e. NTN and $l_{2}$ distance, in the paper.
2. It is pointed that minimizing $\gamma_{i}$, $\gamma_{j}$ and $\||\gamma_{i}-\gamma_{j}\||_{2}$ is the necessary condition of the optimal $\pi$. But since it is not a sufficient condition, more evidence is required to make the approximation justified.

---

> ### Author Response · Authors · 2022-08-02
> **Respond to reviewer HgEJ (Part 1)**
>
> Thank you for your insightful feedback. We answer your questions below.
>
> **Q1:** Analyze the accuracy of this approximation method;
>
> **A1:**  As mentioned in Sec.3, GED measures the difference between two graphs when they are best aligned. In other words, if we perform node permutation on one graph, so that the same-indexed nodes in two graphs have as similar structure as possible, then the GED can be directly computed by computing the difference between the permuted adjacency matrices. In the paper, We refer to the permutation that satisfies such a condition as the optimal permutation $\pi^\star$. However, directly traversing all possible node permutations to find $\pi^\star$ needs complexity $\mathcal{O}(N!)$, which is NP-Hard.  In Sec.3, we rethink this problem in embedding space, and consider necessary conditions that are implicitly required by $\pi^\star$ and that are easy to implement in the model. To that end we define $\gamma_i$, $\gamma_j$ and require both objectives, as well as their difference $\|\|\gamma\_i-\gamma\_j\|\|\_2$ to be minimized. To analyze the accuracy of such approximation, we analyze the sufficiency of these conditions. Notice that $\gamma_i = \sum\_k^N \left\|\left\|\mathrm{DIST}\left(f\_{\theta}(\boldsymbol{A}\_i[k]), g\_{\phi}(\boldsymbol{A}\_i)\right) - \mathrm{DIST}\left(f\_{\theta}(\boldsymbol{A}\_i[k]), g\_{\phi}(\pi(\boldsymbol{A}\_j)) \right) \right\|\right\|\_2$ formally reflects how each node is compatible with the two graphs equally well. A minimum $\gamma_i$ therefore implies the two graphs are properly aligned. Similarly, a minimum $\gamma_j$ achieves the same goal from the standpoint of $G_j$. As an additional perspective, we view $2N$ vectors $\\{f_{\theta}(\boldsymbol{A}\_i[k])\\}^N_{k=1} \cup \\{f_{\theta}(\pi(\boldsymbol{A}\_j)[k])\\}^N_{k=1}$ as a set of anchors, denoted as $\mathcal{T}$. Since $f_\theta(\cdot) \in \mathbb{R}^d$, if the linear span of a subset of  $\mathcal{T}$ can be a space $\mathbb{R}^d$, then for all $k$, a small difference between  $\mathrm{DIST}\left(f_{\theta}(\boldsymbol{A}\_i[k]), g_{\phi}(\boldsymbol{A}\_i)\right)$ and $\mathrm{DIST}\left(f_{\theta}(\boldsymbol{A}\_i[k]), g_{\phi}(\pi(\boldsymbol{A}\_j)) \right)$  can reflect a small distance between $g_{\phi}(\boldsymbol{A}\_i)$ and $g_{\phi}(\pi(\boldsymbol{A}\_j))$, thus we can further get that $\boldsymbol{A}\_i$ and  $\pi(\boldsymbol{A}\_j)$ are very similar input for the injective function, i.e., $\boldsymbol{A}\_i$ and  $\pi(\boldsymbol{A}\_j)$ is best aligned, so we have $\pi = \pi^\star$.  If $\|\|\gamma_i-\gamma_j\|\|\_2$  is minimized meanwhile, it will lead to the distance between $f_{\theta}(\boldsymbol{A}\_i[k])$ and $f_{\theta}(\pi(\boldsymbol{A}\_j)[k])$ takes the minimum, which is the node-level embeddings under $\pi^\star$. Thus, assuming $N > d$, as long as at least $d$ of the $2N$ nodes have different neighbor indexes, there must exist a subset of $\mathcal{T}$ that can span $\mathbb{R}^d$, then our necessary conditions can be sufficient. In most real-world cases the number of nodes $N$ is large than the embedding dimension $d$, and the connected structures of most nodes are not exactly the same due to the complexity of the network. Hence, the sufficiency of our proposed conditions usually can be satisfied in real-world cases.

---

> > ### Comment · Reviewer_HgEJ · 2022-08-08
> > **Thanks for the Response (Part 1)**
> >
> > Thanks for your really detailed and convincing analysis of the approximation accuracy. Now I am more confident about the robustness and effectiveness of ERIC.

---

> ### Author Response · Authors · 2022-08-02
> **Respond to reviewer HgEJ (Part 2)**
>
> **Q2:** Take more types of discriminators into account.
>
> **A2:** In our paper, we use NTN as the base discriminator, and consider 2-order Minkowski distance (i.e., $\ell_2$ distance) as an additional GED discriminator. Experiments show that $\ell_2$ distance and NTN are complementary and combining them as a multi-scale GED discriminator significantly improves the performance compared with using a single discriminator. Here, we still use NTN as the base discriminator, and take higher-order Minkowski distance as additional discriminators such as  $\ell_4$ and $\ell_6$ distance. Besides, we add two variants of ERIC: ERIC(Pearson) and ERIC(HK), which use Pearson coefficient and Heat Kernel as additional discriminators, respectively. We report MSE results for these variants as follows:
>
> |         | ERIC(w/o $\ell_2$) | ERIC      | ERIC ($\ell_4$) | ERIC($\ell_6$) | ERIC(Pearson) | ERIC(HK) |
> | ------- | ------------------ | --------- | --------------- | -------------- | ------------- | -------- |
> | AIDS700 | 1.466              | 1.374     | **1.369**       | 1.387          | 1.409         | 1.431    |
> | LINUX   | 0.253              | **0.107** | 0.117           | 0.114          | 0.152         | 0.137    |
>
> In this Table, ERIC (w/o $\ell_2$) only uses NTN as the single GED discriminator, which achieves the worst performance among all variants. Especially on LINUX, the variants with multiple discriminators improve performance by at least 39.9% relatively compared with ERIC (w/o $\ell_2$).  The results also show that adding only a simple $\ell_2$ discriminator can boost the prediction accuracy. Note that  Minkowski distance, Pearson coefficient and Heat Kernel do not introduce any learnable parameters, thus extending the single GED discriminator to the multi-scale discriminator does not bring an additional training burden.
>
> ## Social Impacts and Limitations:
>
> ERIC can be widely employed in graph similarity search and graph database analysis. For example, in a drug database, ERIC can be used to search for a query chemical compound, in order to find drugs with similar structures and properties. In computer security, we can use ERIC to compute the similarity between communication graphs, which could help identify network intrusions from the graph-based connection records; in social network analysis, the similarity between different user communication graphs can also be computed by ERIC, which can reveal behavioral patterns. ERIC has significant technical-economic and social benefits because it can shorten the labor time and labor intensity of similarity searching. However, to the best of our knowledge, there is not previous work providing GED-based graph datasets with weighted edges. Thus, the performance of ERIC on weighted graph datasets remains unclear. In this regard, we suggest researchers adopt ERIC with caution if their own graph datasets are weighted. We will add the discussion of social impact and limitations in our final version.

---

> > ### Comment · Reviewer_HgEJ · 2022-08-08
> > **Thanks for the Response (Part 2)**
> >
> > Thank you for your detailed feedback. Generally, feedback to my questions is satisfactory, solving some of my questions and unclear misunderstandings. It would be better to add the above discussion to the final version.

---

> > > ### Author Response · Authors · 2022-08-09
> > > **Thanks for your valuable suggestions**
> > >
> > > Thanks for the time to read our feedback. We will include all your suggestions in the final version of the paper.  If you have any further comments and questions, we are glad to write a follow-up response.

---

### Official Review · Reviewer_AeTA · 2022-07-11

**Rating:** 6
**Confidence:** 4
**Soundness:** 3 good
**Presentation:** 3 good
**Contribution:** 3 good

**Summary:**

This paper presents a novel way to compute graph similarity, in particular graph edit distance (GED). The key insight is a proposal of necessary conditions under optimal permutation (i.e., eq. (3) and eq. (4))) that lead to the computation of GED. Based on the insight, a neural model is introduced, critically reducing the computation burden from node-to-node similarity computation (quadratic w.r.t number of nodes) down to node-to-graph similarity computation (linear w.r.t. number of nodes). Experiments validate the effectiveness and efficiency of the proposed method.

**Questions:**

N/A

**Limitations:**

Yes, the authors have adequately addressed the limitations and potential negative societal impact of their work.

**Strengths And Weaknesses:**

Strengths
- Under the proposed necessary conditions in Sec. 3, the computation of GED is reduced to node-to-graph level computation. It greatly reduces the traditional neural methods that take node-to-node level computation.
- The model design is technically reasonable and correct.
- Experiments are well designed and sufficiently show that the proposed methods establish new SOTA results while using significantly less running time. Ablation studies further validate the value of each component.
- The paper is well structured and easy to follow.

Weaknesses: One thing that is unclear is the motivation in Sec.3. Authors propose how they arrive at the necessary conditions for computing GED. However, the derivation involves multiple steps and is somehow conceptual, e.g., quoted from the paper: "a necessary condition is that the distance between $f_θ(A_i [k])$ and $f_θ(\pi^* (A_j )[k])$ is minimized for all $k \in {1, \ldots, N}$." It would be more convincing to include formally mathematical derivations to obtain these necessary conditions.

---

> ### Author Response · Authors · 2022-08-02
> **Respond to reviewer AeTA**
>
> Thanks for the thorough reviews. The main concern lies in the derivation of the necessary conditions under the optimal permutation, which the reviewer found not clear enough. Below we provide a detailed elaboration to address this concern.
>
> GED measures the difference between two graphs under the best structural alignment. However, directly traversing all possible node permutations to obtain the best alignment between $G_i$ and $G_j$ is NP-Hard. We thus consider this question from another perspective: When best aligned, what should the representations of the two graphs look like after embedding?  This question can be answered by deriving the necessary conditions under the optimal permutation. More concretely, given two graphs $G_i$ and $G_j$ with adjacency matrices $\boldsymbol{A}_i$ and $\boldsymbol{A}_j$ of equal size (if not equal, we pad the smaller one with 0). The optimal permutation $\pi^\star$ aims to re-index nodes in one graph, so that same-indexed nodes in two graphs are similar in structure. Necessarily, $\pi^\star$ satisfies:
>
> $$
> \pi^\star = \arg \min_{\pi} \sum_{k}\big|\big| \boldsymbol{A}_i[k] - \pi(\boldsymbol{A}_j)[k] \big|\big|\_2  \tag{A}
> $$
> where $\boldsymbol{A}_i[k] \in \mathbb{R}^N$ is the $k$-th row of $\boldsymbol{A}_i$ , representing the local structure of the node with index $k$ in $G_i$, and $\pi^\star(\boldsymbol{A}_j)$ is the optimal node permutation on $G_j$ . Given a proper injective function $f_\theta (\cdot): \mathbb{R}^N \to \mathbb{R}^d$  that encoders both $\boldsymbol{A}_i[k]$ and $\pi^\star(\boldsymbol{A}_j)[k]$, we expect that the difference between $\boldsymbol{A}_i[k]$ and $\pi^\star(\boldsymbol{A}_j)[k]$ is well reflected by the distance between their embeddings. Hence,  $\pi^\star$ is reformulated as:
>
> $$
> \pi^\star  = \arg \min_{\pi} \sum_{k}\big|\big| f_\theta(\boldsymbol{A}_i[k]) - f_\theta(\pi(\boldsymbol{A}_j)[k]) \big|\big|\_2  \tag{B}
> $$
> This gives us an approximate form of the necessary condition of $\pi^\star$ by virtue of network embedding, as mentioned in Sec.3. Next, we look at the necessary condition of $\pi^\star$ from a global perspective. That is, the distance between the two graphs’ embeddings,  $g_\phi(\boldsymbol{A}_i)$ and $g_\phi(\pi^\star(\boldsymbol{A}_j))$, should be minimal. Since $\sum\_{k}||\boldsymbol{A}_i[k] - \pi(\boldsymbol{A}_j)[k] ||\_2  = ||\boldsymbol{A}_i - \pi(\boldsymbol{A}_j)||\_{21}$, then Eq.(A) can be re-written as:
>
> $$
> \pi^\star = \arg \min_{\pi} \|\|\boldsymbol{A}_i - \pi(\boldsymbol{A}_j)\|\|\_{21}  \tag{C}
> $$
> Given an injective function $g_\phi(\cdot): \mathbb{R}^{N \times N} \to \mathbb{R}^{d}$ to encode $\boldsymbol{A}_i$ and $\pi(\boldsymbol{A}_j)$, the difference between $\boldsymbol{A}_i$ and $\pi(\boldsymbol{A}_j)$ can be approximated by the distance between $g_\phi(\boldsymbol{A}_i)$ and $g_\phi(\pi(\boldsymbol{A}_j))$, giving:
>
> $$
> \pi^\star = \arg \min_{\pi} \|\|g_\phi(\boldsymbol{A}_i) -g_\phi(\pi(\boldsymbol{A}_j))\|\|\_{2}  \tag{D}
> $$
> where we replace  $\ell\_{21}$ with  $\ell\_2$ norm, because for a vector, its $\ell\_{21}$ norm is equal to its $\ell\_2$ norm. Eq.(D) shows that under the optimal permutation $\pi^\star$,  the distance between $g_\phi(\boldsymbol{A}_i)$ and $g_\phi(\pi^\star(\boldsymbol{A}_j))$ takes the minimum values. Thus, the other necessary condition we state in Line 116-117 holds.  Note that  $f_\theta$ and $g_\phi$ are injective, the reason is that we expect the encoders can distinguish different input structures, and in Sec.4 we instantiate them with GIN and DeepSets.
>
> We will take your suggestions into account in the updated version of the paper.

---

### Meta-Review · Area_Chair_ukz9 · 2022-08-27

**Recommendation:** Accept
**Confidence:** Certain

**Metareview:**

Compute similarity between two graphs based on the graph edit distance (GED) is expensive. This paper proposes a learnable way to infer GED quickly. The propose method has two stages, in the first stage, the GNN encoder capture underlying alignment information between pair-wise graphs; in the second stage, it uses the learned encoder to reflect the exact GED value between two graphs.

Questions raised by reviewers, i.e., on motivations /  types of discriminators /  accuracy of this approximation method / sensitivity of lambda, have been addressed during the rebuttal.  Authors are encourage to reflect these changes in the final version.

**Award:**

No

---

### Decision · Program_Chairs · 2022-09-14

Accept